# How research data deliver non-academic impacts: A secondary analysis of UK Research Excellence Framework impact case studies

Eric A. Jensen[1,2]*, Paul Wong[3], Mark S. Reed[4]

1 Institute for Methods Innovation, Trinity Technology & Enterprise Campus, Unit 23, Dublin, Ireland,
2 Department of Sociology, University of Warwick, Coventry, United Kingdom, 3 School of Cybernetics, College of Engineering and Computer Science, Australian National University, Acton, ACT, Australia,
4 Department of Rural Economies, Environment & Society, Thriving Natural Capital Challenge Centre, Scotland's Rural College (SRUC), Edinburgh, United Kingdom

* eric@methodsinnovation.org

**Data Availability Statement:** Computer code is available at https://github.com/squarcleconsulting/ukrefimpactmined (MIT licence). Please note that due to the Term of Use issued by REF UK, the

## Abstract

This study investigates how research data contributes to non-academic impacts using a secondary analysis of high-scoring impact case studies from the UK's Research Excellence Framework (REF). A content analysis was conducted to identify patterns, linking research data and impact. The most prevalent type of research data-driven impact related to "practice" (45%), which included changing how professionals operate, changing organizational culture and improving workplace productivity or outcomes. The second most common category was "government impacts", including reducing government service costs and enhancing government effectiveness or efficiency. Impacts from research data were developed most frequently through "improved institutional processes or methods" (40%) and developing impact via pre-analyzed or curated information in reports (32%), followed by "analytic software or methods" (26%). The analysis found that research data on their own rarely generate impacts. Instead they require analysis, curation, product development or other forms of significant intervention to leverage broader non-academic impacts.

## 1 Introduction

Making a positive difference in the world, or "impact", has long been a driving force for researchers across the disciplinary spectrum, whether they generate research data or not. There is now also growing interest in the impact of research from funders who want evidence of the value of their research investments to society [1, 2]. This interest has been driven, in part, by successive economic crises that have intensified the need to justify continued public investment in research. In response to this, the first national assessment of research impact was conducted by the UK via its Research Excellence Framework in 2014 (REF2014), and now there are national assessments of research impact in The Netherlands, Sweden, Italy, Spain, Norway, Poland, Finland, Hong Kong, Australia, New Zealand and the USA (for details, see Reed et al. [1]). The analysis of impact case studies included in this research is based on the definition of impact used in REF2014, as "an effect on, change or benefit to the economy,

**Funding:** Eric Jensen received funding from the Australian Research Data Common (ARDC) for this study. The ARDC played no role in the design, data collection and analysis of this research.

**Competing interests:** The authors have declared that no competing interests exist.

society, culture, public policy or services, health, the environment or quality of life, beyond academia" [10] (p. 26). According to REF2014 guidance, the primary functions of an impact case study were to articulate and evidence the significance and reach of impacts arising from research beyond academia, clearly demonstrating the contribution that research from a given institution contributed to those impacts [10].

Investment in research data capacity and infrastructure has been a long-standing and important pillar of public funding for research. However, evaluations of the impact of research data have tended to focus on assessments of return on investment. For example, Beagrie and Houghton[3-6] estimated the return on investment for the British Atmospheric Data Centre, the Economic and Social Data Service and the Archaeology Data Service in the UK, and the European Bioinformatics Institute (EMBL-EBI), which was established to manage public life-science data on a large scale. While EMBL-EBI's annual operating cost was £47 million, Beagrie and Houghton [2] estimated that the annual return on investment in using EMBL-EBI data was £920 million with another £1 billion from efficiency gain from using its data and services. Similarly, AuScope is an Australian national infrastructure investment in geoscience and geospatial data capability to support data intensive research and industries. Using similar economic modelling, Lateral Economics estimated that the economic benefits of AuScope was $3912 million AUD, while the total economic cost of AuScope is $261 million AUD, amounting to $15 AUD of benefits for every $1 AUD of investment [3].

Other approaches have also been considered for evaluating the impact of research data, including the use of bibliometrics and altmetrics [4, 5] and usage and end-user surveys [6]. There is clearly a broad interest in understanding how research data and infrastructures contribute to broad socio-economic impacts. This interest is also reflected in the development of an OECD reference framework in 2019 for assessing the scientific and socio-economic impact of research infrastructures [7]. However, each of the metrics used so far to evaluate the impact of research data only provide: a narrow economic perspective on the value of research data (in the case of monetary valuation methods); crude proxies for wider benefits (in the case of bibliometrics and altmetrics); or incomplete assessments of benefits, biased towards the immediate users of data (who are often researchers).

Based on a wider critique of metrics for evaluating impact [8], REF2014 adopted a case study approach, which has now been widely replicated and adapted for impact evaluation internationally (Reed et al. [1]). In the context of REF, case studies are historical narratives that reconstruct past events and claim that research by a submitting institution has had, or is having, causal impacts on society [9]). The use of case studies enabled the integration of metrics with evaluation data generated from a range of other methods (including qualitative), as part of a narrative description of the widest possible impacts arising from research [10]. The resulting publication of over 6,000 impact case studies in 2014 was unique in terms of its size and scope. Although the scores of individual case studies were not made public, it is possible to infer high (3* or 4*) versus low (unclassified, 1* or 2*) from an analysis of the published results. This provides a unique opportunity to conduct secondary analysis of impact case studies that were perceived by evaluation panels to have successfully demonstrated impact (i.e., high scoring cases) to extract new insights about impact generation processes.

In this paper, we have identified a sub-set of case studies that relied heavily on research data, to provide a more comprehensive evaluation of the wider benefits of research data than has ever been possible before. The sample is not representative of UK research institutions or disciplines, and the case studies are self-selected and written to showcase positive impacts rather than to provide more balanced, independent evaluations of outcomes. However, the breadth of case studies in the database makes it possible to evaluate the value of research data across a range of institutions (compared to economic evaluations which tend to focus on

individual institutions), considering data generated and used across the disciplinary spectrum, in an exceptionally wide range of application contexts. Whilst imperfect, the analysis of these case studies offers an opportunity to provide insights into the societal benefits of investing in research data that have not previously been possible. In particular, we seek to: 1) understand the pathways and processes through which research data was used to generate impacts; and 2) evaluate the range of different impacts generated from research data, considering the potential for multiple impacts arising over time in many different domains.

## 2 Methods

The research evaluates the accounts contained in 2014 impact cases for the UK's Research Excellence Framework (REF). The research focuses on the content of REF2014 impact narratives, investigating how research data delivered positive societal outcomes and what factors enabled such outcomes to develop. However, as a linguistic corpus, the detail impact description of all impact cases is made up of over 6,000 distinct cases with 6.3 million words in total. Impact case studies submitted to the exercise followed a set structure: 1 –Summary of the impact; 2 –Underpinning research; 3 –References to the research; 4 –Details of the impact; 5 – Sources to corroborate the impact [10]. Using this structure, case studies had four pages within which to show how a body of work (consisting of up to six research outputs by the submitting institution) contributed, whether directly or indirectly to "an effect on, change or benefit to the economy, society, culture, public policy or services, health, the environment or quality of life, beyond academia". In REF2014, research outputs and impact were peer reviewed in 36 disciplinary 'Units of Assessment'. Panels consisted of academics and research users who were recruited to help evaluate impacts.

As a first step we apply a standard Informal Retrieval (IR) technique (Manning et al. [11], Baeza-Yates and Ribeiro-Neto [12]) to identify candidate cases where data featured prominently in the impact narratives. The aim is to reduce the volume of cases to a manageable size where a quantitative "content analysis" can be used. We focused on cases where the term "data" and its variant "dataset", "datasets", "database" and "databases" are mentioned explicitly in the impact narratives. The underlying procedure has been implemented in R (under MIT license) and is available at https://github.com/squarcleconsulting/ukrefimpactmined.

The application of IR involved creating a simplified representation of the corpus: the impact narratives were converted into a document-term matrix where the Term-Frequency Inverse Document Frequency (TF-IDF) was computed for each term in each impact case. Word occurrences play a fundamental role in this simplified model and the definition of TF-IDF. "Term frequency" (TF) is the ratio between the number of occurrences of a given term in a given impact case and the total number of all term instances in the case. "Inverse Document frequency" (IDF) is the logarithm of the ratio between the total number of cases over the number of cases with at least one occurrence of a given term. TF-IDF is the product of TF and IDF. Intuitively, a word frequently mentioned in an impact case (e.g., "health") can provide information about the case. However, if the word is also mentioned in every impact case, then the word becomes less useful for differentiating the content of different impact cases. By comparing the TF-IDF value of the term "data" (and its variant) against the TF-IDF values of other terms in an impact case, we can provide a rough measure of how prominently the term "data" is featured in the impact narratives. Higher TF-IDF value would mean playing a more prominent role in the narrative. More specifically, the TF-IDF values of all terms in a given impact case were ranked by quartiles ($25^{th}$, $50^{th}$, and $75^{th}$ percentiles). This allowed us to identify 484 impact cases where the TF-IDF values of the term "data" (and its variant) are above the third quartile of the ranking. This is further filtered by focusing on only those cases amongst the 484

cases with a REF rating of 3* or 4* (i.e., those that were high impact rating). There are 148 remaining cases as noted in Table 1. A manual review of these 148 cases were conducted to ensure their relevance.

The second step of this study was conducted using quantitative content analysis, a well-established process for converting unstructured data into structured, numerical data in a reliable and accurate way (Krippendorff [13]; Neuendorf [14]). For this analysis, a subset of the sample (approx. 20%) was used to develop an analytic framework, with specific categories, definitions and examples. The framework was built through manually assessing this subset, extracting relevant parts and refining it over the course of the analysis. This process provided key operational definitions and was designed to address the research questions for the project.

The categories were applied systematically by research assistants independently perusing randomly allocated content. More than one category could be allocated to each impact case study, if applicable. To keep the content analysis focused, impact-related text passages were extracted manually from the longer impact narratives before coding. Finally, the different dimensions uncovered through the quantitative analysis were supplemented by the extraction of case examples. The detailed content analysis codebook (including specific analysis protocols) is available as supplementary material and the full data are available on the open access database Zenodo.

The content analysis process yielded quantifiable results, underpinned by intercoder reliability checks on randomly selected subsets. Intercoder reliability refers to the extent to which independent analysts (or 'coders') evaluating the same content characteristics have reached the same conclusion. A high level of agreement is taken as evidence that the content analysis has identified characteristics that are objectively evident in the texts being analysed (e.g. Jensen & Laurie [15]).

The first step in evaluating inter-coder reliability is to have the members of the coding team independently code the same (randomly selected) sub-set of sample cases. In accordance with this, 10% of the cases analysed by at least two analysts were randomly selected and tested for inter-coder reliability using Krippendorff's Alpha (or 'Kalpha'). As part of this inter-coder reliability randomly selected sub-sample, 52 units were analysed statistically. Variables showing '1.0' in the Kalpha table means that there are no disagreements by the two analysts.

The results (Table 2) show that there were very good inter-coder reliability scores across the variables (all above .8 Kalpha, which is the established benchmark for good reliability). All differences were resolved by the lead researcher.

Finally, it is important to highlight a key methodological limitation in this study. We are using impact case narratives that are being crafted to tell stories about the impact of research. This means that there may be an incentive for case study authors to emphasize how essential the research outputs (and also research data) were to the impacts described in the case. We must be cautious, therefore, in making generalizations to all research-linked impacts that may be taking place, many of which may have been systematically omitted from REF impact case studies specifically because multi-dimensional pathways to impact make causal claims more tenuous.

**Table 1. Different stages of information retrieval to identify candidate cases.**

| Summary | Value |
|---|---|
| Number of impact cases available | 6637 |
| Number of cases with "data" mentioned in impact section | 2213 |
| Number of cases where TF-IDF (data) is above Q3 | 484 |
| Number of cases where TF-IDF (data) is above Q3 and REF rating are 3* or 4* | 148 |

**Table 2. Krippendorff's alpha for variables analysed in this study.**

| Variable Name | Kalpha |
|---|---|
| Impact Type Overall Variable (see Table 3) | 0.9288 |
| Impact Pathways Overall Variable (see Table 5) | 1.0 |
| Impact Pathway: Searchable Database | 1.0 |
| Impact Pathway: Report or Static Information | 0.9618 |
| Impact Pathway: Mobile App | 1.0 |
| Impact Pathway: Analytic Software or Methods | 0.9478 |
| Impact Pathway: Improved Institutional Processes / Methods | 1.0 |
| Impact Pathway: Sharing of Raw Data | 1.0 |
| Impact Pathway: Sharing of Tech / Software | 1.0 |
| Impact Pathway: Other Impact Instrument | 1.0 |
| Impact Pathway: Unclear / Uncertain | 1.0 |

# 3 Results

## 3.1 Types of impact associated with research data

The first level of analysis in this research was designed to identify the types of impacts that are most linked to research data. The initial analysis of the cases revealed a set of different types of impact linked to research data, defined in Table 3.

The most prevalent types of research data-driven impact in our sample were related to *Practice* (45%) and *Government* (21%), which includes both *Government Spending / Efficiency* (6%) and *Other Government / Policy* impacts (15%). Likewise, other types of research data-driven impacts such as *Economic* impact (13%) and *General Public Awareness* impact (10%) were also

**Table 3. Identified types of impact.**

| Impact | Description |
|---|---|
| Government Spending / Efficiency Impact | Reducing the cost of delivering government services; increasing impact/ quality of government service without raising cost. |
| Other Government / Policy Impact | Changing public policy or government regulations, or how either of these are implemented. |
| Practice Impact | Changing the ways that professionals operate; changing organizational culture; improving workplace productivity or outcomes; improving the quality of products or services through better methods, technology, understanding of the problems, etc. |
| General Public Awareness Impact | Improving public knowledge about a topic or increasing public visibility or attention for an issue. |
| Justice / Crime Reduction / Public Safety Impact | Reducing crime; Increasing efficiency in reducing crime; Improving justice outcomes (i.e., fairer; less cost; better social outcomes). |
| Public Health Impact | Improvements to the health of the population or a part of the population. |
| Economic Impact | Greater revenue, profit or market share is developed for a company or sector using research data. |
| Environmental Impact | Improvements in the natural environment, or reductions in threats or harm. |
| Other Kind of General Public Impact | Benefits for the public (not professionals/government) that are not explicitly stated above in another category. |
| Other Non-Academic Impact | REF-eligible non-academic impacts not falling into any of the categories above. (i.e., cannot include academic publications or improvements to teaching within a researcher's own institution, as these would not be REF-eligible). |
| Unclear / Uncertain | Not enough detail or clarity to clearly identify. |

**Table 4. Prevalence of different types of impact associated with research data.**

| Impact Type | Percentage |
|---|---|
| Practice Impact | 45% |
| Other Government / Policy Impact | 15% |
| Economic Impact | 13% |
| General Public Awareness Impact | 10% |
| Government Spending / Efficiency Impact | 6% |
| Public Health Impact | 5% |
| Other Kind of General Public Impact | 3% |
| Remaining | 4% |

represented in a noteworthy minority of cases (Table 4). REF2014 impact case content with impact dimensions that fit in more than one field were categorized for each impact separately.

The findings show that 45% of research data-linked impacts focused on practice (Table 4). In these cases, the research data have been used to develop changes in the ways that professionals operate. These changes have a direct (or indirect) impact on the organisational culture, improving workplace productivity or outcomes or improving the quality of products or services through better methods, technology, understanding of the problems. For example, because of the application of research data collected using the Work-Related Quality of Life (WRQoL) scale, there was a marked improvement in workplace wellbeing documented in an impact case study:

> "*Survey results at organisation level "allowed us to focus on sickness absence", prompting the introduction of a new sickness absence procedure: "days lost [. . .] for stress/mental ill health for academic staff have more than halved from 1453 days to 604 days, a[n annual] saving estimated to be in the region of £100,000".*

Twenty-one percent of impacts were related to government and policy impacts, including at least one of the following benefits:

- Reducing the cost of delivering government services

- Increasing impact/quality of government services, without raising costs

- Changing public policy or government regulations

An example of a case that focuses on reducing the cost of delivering government services or increasing impact/quality of government services without raising costs involved the UK government agency, the Office for National Statistics (ONS). Here, an analysis of research data from the Office for National Statistics was used as a basis to develop a 'disclosure framework and software package' for the release of microdata to a vast range of end-users, including central and local government departments, and the health service. That is, ONS data were used to develop methods and software that in turn could be used to leverage value from ONS data for wide range of stakeholders across government.

Research data were also used to develop impacts on both the economy (13%) and general public awareness (10%). Examples of economic impact included cases that delivered health care cost savings while improving patient health outcomes. General public awareness impacts included changes to how people perceived and understood information related to contemporary topics, such as bullying and social media use, among others. Such general public-oriented

impacts often focused on improving public knowledge about a topic or increasing public visibility or attention for an important issue.

## 3.2 How impact develops from research data

Impacts were developed from research data in several different ways, we refer to as impact pathways. Here, we analyse the nature of these different interventions, which could be understood as the means of developing impact, or the impact generating activities. The impact development approaches we identified are summarized in Table 5.

The most common ways of developing impact from research data were 'Improved Institutional Processes or Methods' (40%), 'Reports or static information' (32%) and 'Analytic Software or Methods' (26%) (Table 6). As multiple ways of developing impact could be used in tandem, the analysis allowed for multiple impact instruments to be identified for a single impact.

Improving 'institutional processes or methods' was a major pathway to developing impact. One example of this category impact development comes from research on regulatory practice:

> "[On regulatory practice] Raab's research has also had a more direct impact on the regulation of information privacy, informing specific policies and frameworks in the UK and beyond. [. . .] His distinct contribution [to the Scottish Government's Identity Management and Privacy Principles report] is evident in principles that warn against discrimination and social exclusion in identification processes; that encourage organisations to raise awareness of privacy issues; and that promote transparency [. . .] These have been widely cited and applied in Scottish Government policy".

Other examples where research data were used to enhance institutional processes include changes in data privacy practices and enhancements in the transparency of companies' operating procedures.

**Table 5. Identified ways of developing impact, or impact pathways.**

| Impact pathways | Description |
| --- | --- |
| Searchable Database | A database that can be accessed to view the research data in a dynamic way (that is, offers the ability to select variables/filters, allowing for customized information to be accessed by users to use for their own purposes). |
| Reports or static information | Report containing pre-analysed/curated information, a static database, results tables or other methods of presenting the research data as processed information to be used without customisation or filtering of the data. |
| Mobile App | An application designed for smartphone or tablet to access the research data or an analysis/results of the data. |
| Analytic Software or Methods | Research data used to generate or refine software or research/analytic methods. |
| Improved Institutional Processes / Methods | Research data used to make an institution's way of operating better/more efficient or more effective at delivering outcomes. |
| Sharing of Raw Data | Research data has an impact via being shared with others (in raw or minimally anonymized form) outside of the research team that generated the data so that they can do something with it (e.g. further analysis, etc.). |
| Sharing of Tech / Software | The research data have an impact via sharing technology or software that was created using the research data or that uses the research data somehow. |
| Other Impact Instrument | A clearly identifiable impact instrument that does not fit into any of the categories listed above. |
| Unclear / Uncertain | Impact instrument that is not detailed enough to clearly place into any pre-specified category. |

**Table 6. Prevalence of impact development pathways.**

| Impact pathways | Percentage |
|---|---|
| Improved Institutional Processes or Methods | 40% |
| Reports (or other static, curated information) | 32% |
| Analytic Software or Methods | 26% |
| Sharing of Tech / Software | 14% |
| Searchable Database | 10% |
| Sharing of Raw Data | 9% |
| Mobile App | 1% |
| Other Impact Pathways | 4% |

Research data were often used to develop impact through the production of 'reports' or other similar types of prepared information. Such reporting distils research data in a way that makes them intelligible for institutions, making the data useful for a wider user base outside of academia.

'Analytic software or methods' were also used to develop impact from research data. An example of this category of impact development can be drawn from the previous example showing a 'government' impact relating to the UK's Office for National Statistics. This is because the way that the researchers developed impact involved using research data to create a "disclosure framework and software package", which was then used to develop impact:

> *"Analysis of research data from the Office for National Statistics used as a basis to develop a 'disclosure framework and software package' for the release of microdata to a vast range of end-users, including central and local government departments, and the health service."*

That is, the mechanism for developing impact in this case was improved methods for data disclosure and new software to implement those improved methods.

## 4 Discussion

The link between data and impact may be explained by a re-purposing of the Knowledge Hierarchy [16, 17] in which data is transformed into information when it is analysed and interpreted, information becomes knowledge when it is understood in a given context, and knowledge becomes "wisdom" or "enlightenment" when it is used effectively or "for good". Definitions of data in the literature emphasise the lack of meaning or value of data that has not been processed, analysed, given context or interpretation [18, 19]. The definitions of wisdom and enlightenment by Zeleny [16] and Ackoff [17] have much in common with definitions of impact, as they suggest that the purpose of data is ultimately to be transformed into something "effective", that "adds value" and is "socially accepted" [20]. Jashapara [21] focuses on the ethical judgement required to use knowledge to "act critically or practically in any given situation (pp. 17–18). It is clear that the perception of whether knowledge has been applied "wisely" will depend on the ontological and epistemological perspective of the evaluator. As Reed et al. [1] suggest, "impact is in the eye of the beholder; a benefit perceived by one group at one time and place may be perceived as harmful or damaging by another group at the same or another time or place".

Such value judgements and assumptions are rarely explicit in definitions of either wisdom or research impact. However, for data to generate impact, it must first be transformed into information and knowledge, and knowledge is inherently subjective. The interpretation of information is strongly influenced by the mode of transmission (e.g. socially mediated via

peers versus mass media; REF), context (e.g. Bandura's idea [22] that all knowledge is formed through "social learning", even when it is not socially mediated, as the interpretation of information is influenced by prevailing social, cultural and political contexts and worldviews) and power relations (e.g. determining who has access to data, how particular interpretations of data are managed to restrict the range of information available, and how information is used to drive decisions through overt and covert forms of power over and power with [23, 24]. In this light, Reed et al. [1] define research impact as, "demonstrable *and/or perceived* benefits to individuals, groups, organisations and society (including human and non-human entities in the present and future) that are causally linked (necessarily or sufficiently) to research" (emphasis added).

The research findings pertaining to impact pathways show how data from multiple disciplines contributed towards the generation of impact and emphasizes the importance of both analysing and interpreting data, and making this information easily accessible to decision-makers. Indeed, 'reports or static information' (32%) and 'analytic software or methods' (26%) were among the most frequent ways in which REF impact case studies claimed impacts arose from data. It is likely that these reports, software and methods played an important role in establishing causal relationships between research and impact, which has been shown to be a strong predictor of high scores in REF2014 case studies (Reichard et al. [25]). Reichard et al. found that high scoring case studies contained more attributional phrases connecting research to impact, and contained more explicit causal connections between ideas, and more logical connectives than low-scoring cases. The most common way in which data led to impact was via 'improved institutional processes or methods' (40%), suggesting the importance of adaptively (re-)interpreting information in specific operational contexts. This is consistent with the finding of Bonaccorsi et al. [9] that institutional agents (including university, government, NHS and the European Union) played an important role in demonstrating claims that research was causally related to impacts in REF2014 case studies (especially in the social sciences and humanities, which cited more agents per case study than STEM subjects). Louder et al. [26] have argued that it is essential for such process-related impacts to be captured in case studies, and Gow and Redwood [27] contend that they should have an impact category of their own, namely "procedural impacts".

This study is the first attempt to analyse the development of impact from data, focusing for the first time on examples of research where data played a major role in the generation of both research findings and impact. It suggests that analysis, curation, product development or other strong interventions are needed to leverage value from research data. These interventions help to bridge the gap between research data and potential users or beneficiaries. While good data management, open data and streamlined access to data are necessary, further interventions are needed to maximize impact from research data. Extending the use of research data beyond the scope of academia requires not only traditional academic research skills, but also capabilities in public communication, entrepreneurship and boundary-crossing [27]. The findings show how impact can be facilitated through closer links between government, industry and researchers, capacity building and funding both for researchers to effectively use research data for developing impact and for potential beneficiaries to establish links with researchers and make research data available in usable formats.

## Acknowledgments

We would like to express deep appreciation for the time, consideration and input of the ARDC Advisory Group. The project also benefited from the contributions of the UK REF's impact case study authors and the open publication of these case studies. Finally, researchers

contributing to this project include: Dr Jessica Norberto, Dr Benjamin Smith, Dr Aaron Jensen, Lars Lorenz, Christian Moll.

## Data and code availability

The underlying computer code and data for this research are available at

- https://github.com/squarcleconsulting/ukrefimpactmined (MIT licence)

- https://doi.org/10.5281/zenodo.3543505 (CC-BY 4.0 International Attribution)

- This work has benefited from a large number of R packages, including R core language version 3.4.4 [28], dplyr [29], easypackages [30], geometry [31], gglot2 [32], ggpubr [33], hugh [34], igraph [35], LDAvis [36], mallet [37], NLP [38], readr [39], refimpact [40], rJava [41], rsvd [42], Rtsne [43–45] stm [46], stmCorrViz [47], stringr [48], textminR [49], tidyr [50], tidytext [51], tm [52, 53], and topicmodels [54].

## Author Contributions

**Conceptualization:** Eric A. Jensen, Paul Wong, Mark S. Reed.

**Data curation:** Eric A. Jensen, Paul Wong.

**Formal analysis:** Eric A. Jensen, Paul Wong.

**Investigation:** Eric A. Jensen, Paul Wong, Mark S. Reed.

**Methodology:** Eric A. Jensen.

**Software:** Paul Wong.

**Writing – original draft:** Eric A. Jensen.

**Writing – review & editing:** Eric A. Jensen, Paul Wong, Mark S. Reed.

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
