## [Decision Letter · Decision Letter 0]

24 Feb 2021

PONE-D-20-30429

How research data delivers non-academic impacts: A secondary analysis of UK Research Excellence Framework impact case studies

PLOS ONE

Dear Dr. Wong,

Thank you for submitting your manuscript to PLOS ONE. After careful consideration, we feel that it has merit but does not fully meet PLOS ONE’s publication criteria as it currently stands. Therefore, we invite you to submit a revised version of the manuscript that addresses the points raised during the review process. Please submit your revised manuscript by Apr 10 2021 11:59PM. If you will need more time than this to complete your revisions, please reply to this message or contact the journal office at plosone@plos.org. Please include the following items when submitting your revised manuscript:

We look forward to receiving your revised manuscript.

Kind regards,

Lutz Bornmann

Academic Editor

PLOS ONE

Journal Requirements:

3. We note you have included a table to which you do not refer in the text of your manuscript. Please ensure that you refer to Table 1, 5 and 6 in your text; if accepted, production will need this reference to link the reader to the Table.

Reviewers' comments:

Reviewer's Responses to Questions

**Comments to the Author**

1. Is the manuscript technically sound, and do the data support the conclusions?

Reviewer #1: Partly

Reviewer #2: Partly

2. Has the statistical analysis been performed appropriately and rigorously? 

Reviewer #1: Yes

Reviewer #2: Yes

3. Have the authors made all data underlying the findings in their manuscript fully available?

Reviewer #1: Yes

Reviewer #2: Yes

4. Is the manuscript presented in an intelligible fashion and written in standard English?

Reviewer #1: Yes

Reviewer #2: Yes

5. Review Comments to the Author

Reviewer #1: The manuscript "How research data delivers non-academic impacts: A secondary analysis of UK Research Excellence Framework impact case studies" analyzes the different kinds of non-academic impact, research data from impact case studies from the REF have had.

Overall, the manuscript should be of interest to the readers of PLoS One. The applied methodology seems to be valid. However, some revisions should be made before publication.

The existing literature about non-academic impact in relation to REF should be discussed. A quick search in Web of Science [ ts=(REF and ("societal impact" or "non-academic impact") ) ] reveals previous studies that would be suitable for inclusion in such a short literature overview. The DOIs of 13 out of the 14 papers are: 10.1186/s13063-020-04425-9, 10.1093/bjc/azz076, 10.1007/s11024-019-09384-3, 10.1080/03075079.2018.1455082, 10.1002/berj.3572, 10.1093/reseval/rvz015, 10.1016/j.joi.2019.01.008, 10.1007/978-3-319-95723-4_4, 10.1371/journal.pone.0173152, 10.1007/s11192-016-2115-y, 10.1177/1748048516655718, 10.1093/reseval/rvv007, and 10.1093/reseval/rvu028.

Lines 159 and 160 refer to Krippendorff’s Alpha as "Kalpha". However, Table 2 uses "Alpha" as a column header. That is somewhat confusing.

Lines 181-183 contain duplicated phrases ("... types of impacts that are most commonly linked most commonly linked to research data. ... revealed a set of different types of impact different types of impact ...") and do not have a proper ending. This paragraph has to be rephrased.

Some table numbers seem to be wrong. Please check the cross-referencing in lines 192, 196, 233, and 238.

Table 3 defines eleven types of impact but Table 4 contains only seven types of impact plus a type named "other". Are the remaining types grouped into "other"? If so, this should be explained in the text. This new group "other" seems rather redundant as there has already been another group with "Other Kind of General Public Impact"

Line 201 has a duplicated explanation of the abbreviation WRQoL, once in line and once in the footnote. One explanation is enough.

On Table 6, the impact developments "Mobile App" and "Unclear/Uncertain are missing in comparison with Table 5 without any explanation.

After adding the short literature overview mentioned above, the authors should highlight their contributions beyond the knowledge from prior literature.

I appreciate that the authors are providing data and code. However, the manuscript should include more information about the used program, i.e., which version of R was used, and which packages were loaded? The code loads a long list of packages. These packages have recommended citations that should be included in the manuscript, see for example: citation('ggplot2').

Reviewer #2: This paper examines case studies provided by the Research Excellence Framework (REF). In particular, the paper focuses on highly data-related research activities and distinguishes different types of impact these research activities generate and by which means this is accomplished. This is a highly relevant topic and the idea to use text analyses based on the REF data seems to be promising. However, there are some points (mostly on the methodological approach and how it is described) that should be addressed in a revised version:

The general research question and how the methodological approach is aligned to the research question seems a bit unclear: is it about (1) the impact of providing research data, or (2) the impact of using research data (or something else). The methodological approach does not seem to be able to distinguish these two approaches. If (2) should be the focus of this paper, I am not sure whether focusing on the TF-IDF values above the 3rd quartile (ll. 140-143) is necessary at all (if "data" is mentioned, the case is very likely to make use of some sort of data; what does it mean if the term "data" is mentioned many times in a submission, compared to only once?).

98: "3" -> "3*"

Section 2: A more detailed description of the REF data would be helpful, e.g.: What entities are evaluated in the REF2014 (Papers, Projects, …)? Who evaluates them? What do the *-ratings mean? From which disciplines are the cases / how does the distribution across disciplines look like (if possible)? How long are the narratives / is there a minimum/maximum number of words? This could be placed in an additional subsection in section 2. Also the paragraph from line 167 to line 173 could be moved in this new subsection.

127-129: what does "are … featured prominently" mean here, can this be specified (e.g., if the share of words indicating data-related case exceeds a certain threshold, the case is selected)?

140-142: "… are within the third quartile": Should this be "… are above the third quartile"? The third quartile is a particular number, so a TF-IDF cannot be within the third quartile.

141-143: For the cases identified as data-related, a manual check on what role research data actually play in these cases could be included here. This may also help to address my first comment.

144: Here, the authors could mention (in the text) the number of cases remaining after excluding all cases not having 3* or 4* ratings, and also insert a reference to table 1.

147-151: A short description of the content analysis (how the categories were defined, how they were assigned to the cases) would help the reader to understand the results and also reliability analysis.

159: Are the 52 units all of the units that occur in the data?

Table 2: Where are these categories used? They seem to be similar to the categories presented in Table 3 and Table 5, but they are not identical.

181-183: There seem to be some typos in this paragraph.

Section 3.1: Some additional information (and a reference to the supplementary material) on the impact types presented in the table and how they were defined / assigned to the cases would be helpful to understand the result and to assess their objectivity. In particular, the following questions should be answered: The categories in Table 3 don't occur in Table 2; are Why are the two categories "Government Spending / Efficiency Impact" and "Other Government / Policy Impact" listed separately in the table, while they are merged in lines 189-190? A problem of these impact types may be that they have different levels of specificity (e.g., "Practice Impact" is a very general impact type that may apply to cases from various disciplines, i.e. it is no surprise that it is assigned to many cases; in contrast, "Public Health Impact" or "Environmental Impact" are more specific, since they refer to a particular discipline).

188-192: Can multiple impact types be assigned to one case, or exactly one impact type for each case?

192 & 196: "Table 2" -> should this be Table 4?

201-202: "Work-Related Quality of Life Scale(WRQoL) scale" -> "Work-Related Quality of Life (WRQoL) scale"

Section 3.2: As in section 3.1, some information on how the categories presented in Table 5 were determined and assigned to the cases would be helpful to understand (see the comments on section 3.1). Also the specificity of the categories seems also be problematic here and should be addressed.

238: "Table 4" -> "Table 6"?

302-303: "… emphasises the importance of both analysing and interpreting data, and making this information easily accessible to decision-makers" -> I would agree that these are important issues. But I don't see how this paper backs this up. Instead, the paper focuses on which type of impact is generated and by which means this is done (which is already a highly relevant topic). Thus, I would suggest that this sentence is reformulated accordingly (either the second part removed, or some additional arguments given backing up this claim).

6. PLOS authors have the option to publish the peer review history of their article (what does this mean?). If published, this will include your full peer review and any attached files.

Reviewer #1: No

Reviewer #2: No

---

## [Author Response · Author response to Decision Letter 0]

31 Jul 2021

Please see Response to Reviewers letter for point by point responses to comments.

---

## [Decision Letter · Decision Letter 1]

7 Oct 2021

PONE-D-20-30429R1How research data deliver non-academic impacts: A secondary analysis of UK Research Excellence Framework impact case studiesPLOS ONE

Dear Dr. Wong,

Thank you for submitting your manuscript to PLOS ONE. After careful consideration, we feel that it has merit but does not fully meet PLOS ONE’s publication criteria as it currently stands. Therefore, we invite you to submit a revised version of the manuscript that addresses the points raised during the review process.  Please submit your revised manuscript by Nov 21 2021 11:59PM. If you will need more time than this to complete your revisions, please reply to this message or contact the journal office at plosone@plos.org. Please include the following items when submitting your revised manuscript:A rebuttal letter that responds to each point raised by the academic editor and reviewer(s). You should upload this letter as a separate file labeled 'Response to Reviewers'.A marked-up copy of your manuscript that highlights changes made to the original version. You should upload this as a separate file labeled 'Revised Manuscript with Track Changes'.An unmarked version of your revised paper without tracked changes. You should upload this as a separate file labeled 'Manuscript'.

We look forward to receiving your revised manuscript.

Kind regards,

Lutz Bornmann

Academic Editor

PLOS ONE

Reviewers' comments:

Reviewer's Responses to Questions

**Comments to the Author**

1. If the authors have adequately addressed your comments raised in a previous round of review and you feel that this manuscript is now acceptable for publication, you may indicate that here to bypass the “Comments to the Author” section, enter your conflict of interest statement in the “Confidential to Editor” section, and submit your "Accept" recommendation.

Reviewer #1: (No Response)

Reviewer #2: (No Response)

2. Is the manuscript technically sound, and do the data support the conclusions?

Reviewer #1: Yes

Reviewer #2: Yes

3. Has the statistical analysis been performed appropriately and rigorously? 

Reviewer #1: Yes

Reviewer #2: Yes

4. Have the authors made all data underlying the findings in their manuscript fully available?

Reviewer #1: No

Reviewer #2: Yes

5. Is the manuscript presented in an intelligible fashion and written in standard English?

Reviewer #1: Yes

Reviewer #2: Yes

6. Review Comments to the Author

Reviewer #1: Previous literature about non-academic impact in relation to REF should be reviewed more inclusively in the introduction.

Table 2 contains main empty rows that probably should be deleted.

I tried to reproduce the results. I obtain an error: 'inputs/ImpactOnly.csv' does not exist in current working directory'.

The error seems to occur in line 96 of the R script refimpact.R:

ImpactCase <<- read_delim("inputs/ImpactOnly.csv", "\\t", escape_double = FALSE, trim_ws = TRUE)

I find only the following files in that directory:

0-GPA.csv CaseStudyID.csv impactid-ukprn.csv UoA.csv

Please check the Git repository.

Reviewer #2: The paper has much improved and all points have been adressed. There are only a few minor points that should be changed/clarified:

Table 1: "in Q3" -> "above Q3"

Table 2: This table is much more understandable now, but now the different dimensions for impact type are missing, while the dimensions for impact pathway are still included. Showing the results also for the impact pathway dimensions, or giving a reason why you don't show them, would make this more consistent.

257-258: "Table 5. Identified ways of developing impact, or impact pathways." -> "Table 5."

7. PLOS authors have the option to publish the peer review history of their article (what does this mean?). If published, this will include your full peer review and any attached files.

Reviewer #1: No

Reviewer #2: No

---

## [Author Response · Author response to Decision Letter 1]

10 Oct 2021

Reviewer #1: Previous literature about non-academic impact in relation to REF should be reviewed more inclusively in the introduction.

As noted previously additional literature has been integrated into the introduction and discussion, focusing on other recent analyses of REF2014 case studies, and the inclusion of impact in REF. We would appreciate the reviewer point us to specific papers or publications that should be included for review.

Table 2 contains main empty rows that probably should be deleted.

Empty rows deleted. Please note Word does not track changes to modification of table structure.

Reviewer #1

I tried to reproduce the results. I obtain an error: 'inputs/ImpactOnly.csv' does not exist in current working directory'.

The error seems to occur in line 96 of the R script refimpact.R:

ImpactCase <<- read_delim("inputs/ImpactOnly.csv", "\\t", escape_double = FALSE, trim_ws = TRUE)

I find only the following files in that directory:

0-GPA.csv CaseStudyID.csv impactid-ukprn.csv UoA.csv

Please check the Git repository.

We very much appreciate the reviewer taking the effort to rerun the R script. Line 96 does reference a file called ImpactOnly.csv. But the omission of ImpactOnly.csv is intentional. The universities in the UK could not agree on a single licensing arrangement. As such many universities retained their rights over their impact cases. While we have obtained permission from the REF UK to use the impact cases for our study here, we do not have the right to make copies and redistribute these cases (see Term of Use, section on Copies for research). As such, we have not included the file ImpactOnly.csv in the Git repository – the file contains the entire collection of impact description of cases and redistribution of this file could be construed as a violation of the Term of Use. Having said this, we did provide instructions on how to obtain the collection. The README.md file included in the code distribution, section on “Download the Whole Collection” states:

By default, refimpact.R assumes that the user has manually downloaded the entire collection and saved the file in ukrefimpact/inputs/ImpactOnly.csv. Follow the instruction below to download the whole collection:

• Go to UK Impact Case Studies and click “See all case studies”

• Scroll down and click “None selected”

• Check “Details of impact” and proceed to click “download”

• When download is completed, open CaseStudies.xlsx in Excel

• Rename columns: “Case Study Id” with “ID”, “Unit of Assessment” with “FoR” and “Details of the impact” with “impact”

• Use TRIM(CLEAN(SUBSTITUTE(cell, CHAR(160), “ “))) in Excel to remove all non-printable characters (including line breaks) and extra white spaces in “impact” column

• Remove column “Institution” and “Title”

• Save CaseStudies.xlsx as a tab delimited text file with the name “ImpactOnly.csv”

• Put ImpactOnly.csv in refimpact/inputs

Alternatively, you can turn on automatic download with the fetchUKImpact(auto=TRUE) call. But by default the function will only download the first five impact cases, you can set the value using k=... option.

Please also note that the README.md also states that the R version tested was 3.4.4 under Ubuntu 16.04.5 and OSX 10.13.6. We cannot guarantee the code will run under a different operating systems or environment. 

Reviewer #2: The paper has much improved and all points have been addressed. There are only a few minor points that should be changed/clarified:

Table 1: "in Q3" -> "above Q3"

Amended.

Table 2: This table is much more understandable now, but now the different dimensions for impact type are missing, while the dimensions for impact pathway are still included. Showing the results also for the impact pathway dimensions, or giving a reason why you don't show them, would make this more consistent.

Table 2 has referenced both table 3 and 5. We feel that the inclusion of details from table 3 and 5 inside table 2 would make it more difficult to interpret and understand table 2.

257-258: "Table 5. Identified ways of developing impact, or impact pathways." -> "Table 5."

It is unclear to us why the title of table 5 should be omitted. All the tables in the paper include a short title/description. Such an omission would make it inconsistent with the rest of the paper.

---

## [Decision Letter · Decision Letter 2]

17 Nov 2021

PONE-D-20-30429R2How research data deliver non-academic impacts: A secondary analysis of UK Research Excellence Framework impact case studiesPLOS ONE

Dear Dr. Wong,

Thank you for submitting your manuscript to PLOS ONE. After careful consideration, we feel that it has merit but does not fully meet PLOS ONE’s publication criteria as it currently stands. Therefore, we invite you to submit a revised version of the manuscript that addresses the points raised during the review process. Please submit your revised manuscript by Jan 01 2022 11:59PM. If you will need more time than this to complete your revisions, please reply to this message or contact the journal office at plosone@plos.org. Please include the following items when submitting your revised manuscript:A rebuttal letter that responds to each point raised by the academic editor and reviewer(s). You should upload this letter as a separate file labeled 'Response to Reviewers'.A marked-up copy of your manuscript that highlights changes made to the original version. You should upload this as a separate file labeled 'Revised Manuscript with Track Changes'.An unmarked version of your revised paper without tracked changes. You should upload this as a separate file labeled 'Manuscript'.If applicable, we recommend that you deposit your laboratory protocols in protocols.io to enhance the reproducibility of your results. Protocols.io assigns your protocol its own identifier (DOI) so that it can be cited independently in the future. For instructions see: https://journals.plos.org/plosone/s/submission-guidelines#loc-laboratory-protocols. Additionally, PLOS ONE offers an option for publishing peer-reviewed Lab Protocol articles, which describe protocols hosted on protocols.io. Read more information on sharing protocols at https://plos.org/protocols?utm_medium=editorial-email&utm_source=authorletters&utm_campaign=protocols.

We look forward to receiving your revised manuscript.

Kind regards,

Lutz Bornmann

Academic Editor

PLOS ONE

Journal Requirements:

Reviewers' comments:

Reviewer's Responses to Questions

**Comments to the Author**

1. If the authors have adequately addressed your comments raised in a previous round of review and you feel that this manuscript is now acceptable for publication, you may indicate that here to bypass the “Comments to the Author” section, enter your conflict of interest statement in the “Confidential to Editor” section, and submit your "Accept" recommendation.

Reviewer #1: All comments have been addressed

Reviewer #2: (No Response)

2. Is the manuscript technically sound, and do the data support the conclusions?

Reviewer #1: Yes

Reviewer #2: Yes

3. Has the statistical analysis been performed appropriately and rigorously? 

Reviewer #1: Yes

Reviewer #2: Yes

4. Have the authors made all data underlying the findings in their manuscript fully available?

Reviewer #1: Yes

Reviewer #2: Yes

5. Is the manuscript presented in an intelligible fashion and written in standard English?

Reviewer #1: Yes

Reviewer #2: Yes

6. Review Comments to the Author

Reviewer #1: (No Response)

Reviewer #2: There is only one minor point I would still recommend to change. The title of table 5 should be removed in the text (ll. 257-258), not in the table caption itself (l. 260). All other mentions of tables in the text do not seem to include the titles (e.g. "There are 148 remaining cases as noted in Table 1. A manual review of these 148 cases were conducted to ensure their relevance", ll. 155-157).

7. PLOS authors have the option to publish the peer review history of their article (what does this mean?). If published, this will include your full peer review and any attached files.

Reviewer #1: No

Reviewer #2: No

---

## [Author Response · Author response to Decision Letter 2]

16 Jan 2022

See upload response to reviewer and editor comments.

---

## [Decision Letter · Decision Letter 3]

22 Feb 2022

How research data deliver non-academic impacts: A secondary analysis of UK Research Excellence Framework impact case studies

PONE-D-20-30429R3

Dear Dr. Wong,

We’re pleased to inform you that your manuscript has been judged scientifically suitable for publication and will be formally accepted for publication once it meets all outstanding technical requirements.

Kind regards,

Lutz Bornmann

Academic Editor

PLOS ONE

Additional Editor Comments (optional):

Reviewers' comments:

Reviewer's Responses to Questions

**Comments to the Author**

1. If the authors have adequately addressed your comments raised in a previous round of review and you feel that this manuscript is now acceptable for publication, you may indicate that here to bypass the “Comments to the Author” section, enter your conflict of interest statement in the “Confidential to Editor” section, and submit your "Accept" recommendation.

Reviewer #2: (No Response)

2. Is the manuscript technically sound, and do the data support the conclusions?

Reviewer #2: Yes

3. Has the statistical analysis been performed appropriately and rigorously? 

Reviewer #2: Yes

4. Have the authors made all data underlying the findings in their manuscript fully available?

Reviewer #2: Yes

5. Is the manuscript presented in an intelligible fashion and written in standard English?

Reviewer #2: Yes

6. Review Comments to the Author

Reviewer #2: In the latest version of the document, it seems the title of table 5 is still included in the reference to this table in l. 257. Since this is only a minor formatting issue, I recommend to accept the manuscript for publication, but you may want to check this before publication.

7. PLOS authors have the option to publish the peer review history of their article (what does this mean?). If published, this will include your full peer review and any attached files.

Reviewer #2: No

---

## [Editor Report · Acceptance letter]

1 Mar 2022

PONE-D-20-30429R3 

How research data deliver non-academic impacts: A secondary analysis of UK Research Excellence Framework impact case studies 

Dear Dr. Wong:

I'm pleased to inform you that your manuscript has been deemed suitable for publication in PLOS ONE. Congratulations! Your manuscript is now with our production department. 

Kind regards, 

on behalf of

Dr. Lutz Bornmann 

Academic Editor

PLOS ONE